# Association between hemoglobin/red blood cell distribution width ratio and acute kidney injury in sepsis and heart failure patients

Petherson Mendonça dos Santos[1*◉], Isabele Pardo[2◉],
Maria Carolina Borges Pereira de Almeida[2], Rafael Santana de Oliveira[2],
Fernanda Gabas Miglioli[2], Beatriz Mota Busnardo[2], Felipe Prieto Siqueira[2],
Daniela Harsanyi[3], Andreia Pardini[3], Bruno Bravim[3],
Anna Carolina de Rizzo Cantoni Rosa[4], Kirliane de Sousa Rodrigues Lacerda[5],
Yvve Priscila Gatto[6], Lucas Andrade Pinheiro[7], Rayane Alves Moreira[8,9],
Mateus Américo Galvão Santos[10], João Paulo Fonseca da Silva[9],
João Marcos Santos da Rocha[11], Remo Holanda M. Furtado[12],
Beatriz Moreira Silva[1◉], Miguel Angelo Goes[1,2,3◉]

1 Division of Nephrology, Escola Paulista de Medicina/ Federal University of São Paulo, São Paulo, São Paulo, Brazil, 2 Medical Department, Faculdade Israelita de Ciências da Saúde Albert Einstein/ Hospital Israelita Albert Einstein, São Paulo, Brazil, 3 Department of Medicine, Escola Paulista de Medicina/ Federal University of São Paulo, São Paulo, São Paulo, Brazil, 4 Medical Department, Universidade Santo Amaro, São Paulo, Brazil, 5 Medical Department, Centro Universitário Euro-Americano, Brasília, Distrito Federal, Brazil, 6 Department of Anesthesiology, Unidade de Anestesiologia e Medicina Perioperatória, Hospital Regional de Ceilândia, Brasília, Distrito Federal, Brazil, 7 Emergency Department, Hospital Santa Lúcia Sul, Brasília, Distrito Federal, Brazil, 8 Nursing Department, Programa de Pós-Graduação em Enfermagem, Universidade de Brasília, Brasília, Distrito Federal, Brazil, 9 Nursing Department, Escola de Saúde Pública do Distrito Federal, Fundação de Ensino e Pesquisa em Ciências da Saúde, Brasília, Distrito Federal, Brazil, 10 Department of Rehabilitation and Palliative Care, Hospital Total Health, Brasília, Distrito Federal, Brazil, 11 Nursing Department, Faculdade de Ciências e Tecnologias em Saúde, Universidade de Brasília, Brasília, Distrito Federal, Brazil, 12 Department of Cardiology, Hospital Sírio Libanês, São Paulo, São Paulo, Brazil

◉ These authors contributed equally to this work.
* pethersonmds@outlook.com

## Abstract

### Introduction

The hemoglobin/red blood cell distribution width (Hb/RDW) ratio has emerged as a potential biomarker for acute kidney injury (AKI), particularly in patients with cardio-vascular conditions. This study investigated the relationship between Hb/RDW ratio and AKI incidence in critically ill patients diagnosed with sepsis and heart failure (HF).

### Methods

A retrospective study was conducted with 119 critically ill patients with sepsis and 83 patients with HF, analyzed according to the presence or absence of kidney injury. Multivariable logistic regression identified independent predictors of AKI. Outcomes between higher and lower Hb/RDW groups were compared.

**Data availability statement:** All relevant data are within the manuscript and its Supporting Information files. Additionally, the dataset supporting the findings of this study is available from the Figshare repository (DOI: https://doi.org/10.6084/m9.figshare.28200248) and is openly accessible.

**Funding:** The author(s) received no specific funding for this work.

**Competing interests:** The authors have declared that no competing interests exist.

## Results

Patients who developed AKI showed higher C-reactive protein levels, elevated RDW ($15.7 \pm 2.2$ vs. $14.9 \pm 1.8$; $p = 0.01$), and higher SAPS 3 scores, along with markedly lower Hb concentrations and Hb/RDW ratios ($75.1 \pm 1.6$ vs. $85.5 \pm 1.9$; $p < 0.001$). In multivariable analysis, serum urea (OR 1.016; 95% CI 1.005–1.027 per mg/dL), SAPS 3, and Hb/RDW ratio (OR 0.977; 95% CI 0.959–0.996) were independently associated with AKI. Patients with a lower Hb/RDW ratio had higher frequencies of AKI, kidney-replacement therapy, red-cell transfusion, and mortality. During a 7-year follow-up, progression to dialysis-dependent stage V chronic kidney disease (CKD-V) occurred in 10.8% of HF patients and 2.5% of sepsis patients, indicating that a lower Hb/RDW ratio was also associated with worse long-term renal outcomes.

## Conclusion

The Hb/RDW ratio is independently associated with AKI and may also reflect long-term kidney prognosis, representing a cost-effective and readily available ICU marker to identify patients at risk for both acute and chronic renal deterioration in sepsis or HF.

## Introduction

Acute kidney injury (AKI) and anemia are two prevalent clinical conditions that can significantly impact the outcomes of critically ill patients [1–4]. AKI is a complex syndrome characterized by a rapid reduction in kidney function and is most commonly associated with sepsis (S-AKI) in critically ill patients [5,6]. Additionally, AKI is a frequent occurrence in patients with acutely decompensated heart failure (ADHF), where it is classified as type 1 cardiorenal syndrome (T1-CRS) [7].

Anemia, which often occurs in critically ill patients, can exacerbate clinical conditions, particularly in patients with sepsis [8]. Its impact on the outcomes of acutely ill patients cannot be underestimated [9]. In patients with ADHF, anemia may lead to decompensation, reduced quality of life, impaired oxygen transport, and a worsened prognosis [10].

Red blood cell distribution width (RDW) is a potential biomarker that reflects the degree of anisocytosis characterized by red blood cells of varying sizes. RDW is also helpful in the differential diagnosis of anemia [11,12]. Elevated RDW often results from ineffective red blood cell (RBC) production or excessive cell fragmentation [13]. In critically ill patients with AKI, high RDW levels are not only a marker but also an independent risk factor for all-cause mortality in the intensive care unit (ICU) [14]. Additionally, RDW may serve as a biomarker for predicting outcomes in adult sepsis patients [4,15]. Furthermore, a lower hemoglobin/red blood cell distribution width ratio (Hb/RDW), which reflects the balance between anemia and anisocytosis, has been associated with a higher risk of adverse outcomes in patients with ADHF [16].

Given the exploratory and hypothesis-generating nature of this investigation, we designed this study as a proof-of-concept cohort to evaluate the potential prognostic value of the Hb/RDW ratio for AKI in patients with sepsis and ADHF.

This study investigates the relationship between anemia and AKI to enhance risk stratification for patients with S-AKI and T1-CRS clinical phenotypes. We compared Hb concentration, RDW, Hb/RDW ratio, kidney function, and clinical and demographic data between AKI and non-AKI groups of critically ill patients with sepsis and ADHF. By analyzing clinical data related to the Hb/RDW ratio, we aim to gain insights into the outcomes of phenotypes of critically ill patients with sepsis and ADHF. This research is vital for improving patient care and recovery strategies.

## Methods

### Study population, clinical definitions, and data collection

This is a retrospective cohort study conducted with patients from Einstein Hospital Israelita in São Paulo, Brazil. The study analyzes kidney outcomes of patients with sepsis and ADHF admitted to the ICU from January 1st to December 31st, 2017. All consecutive ICU admissions meeting diagnostic criteria for sepsis or acute decompensated heart failure (ADHF) during this fixed one-year period were screened for eligibility. A total of 386 critically ill adults were initially identified. After applying predefined exclusion criteria, 202 patients were included in the final cohort.

We accessed and collected patients' clinical data from the electronic medical record for the purposes of this research between January 26, 2019 to June 30, 2024, with approval from the local research ethics committee. Moreover, we conducted follow-up assessments for up to seven years to evaluate long-term renal outcomes, specifically the progression to dialysis-dependent stage V chronic kidney disease (CKD-V). Data were analyzed up to July 30, 2024.

We studied 119 patients with sepsis and 83 patients with ADHF. Inclusion criteria encompassed individuals over 18 years old admitted to the ICU with a diagnosis of ADHF or patients with a diagnosis of sepsis. Exclusion criteria comprised dialysis-dependent 5-stage chronic kidney disease, Hematological diseases such as leukemias and lymphomas, oncological diseases of solid organs, cirrhosis, chronic human immunodeficiency virus (HIV) infection, Hepatitis B and C infections, and those who died within the first 48 hours of ICU stay.

Patients in the ADHF group were comprehensively screened using medical records including history, physical examination, and transthoracic echocardiogram results [17].

Patients in the sepsis group exhibited early signs of low perfusion with organ dysfunction, such as decreased blood pressure, drowsiness or mental confusion, reduced diuresis, declining platelet counts, blood coagulation abnormalities, respiratory disorders, or elevated arterial lactate [18,19].

Upon ICU admission, we collected several variables, including: age, sex at birth, mean arterial pressure (MAP), hemoglobin (Hb) concentration, mean corpuscular volume, red cell distribution width (RDW), platelet count, white blood cell count, serum sodium and potassium levels, total bilirubin, arterial blood gas parameters, lactate levels, serum creatinine (sCr), urea, glucose, and N-terminal B-type natriuretic peptide (NT-proBNP). Additionally, we calculated the Simplified Acute Physiology Score 3 (SAPS3) prognostic index [20].

We also assessed the ratio of arterial oxygen pressure to inspired oxygen fraction ($PaO_2/FiO_2$) and the Hb/RDW ratio. Furthermore, we monitored key outcomes during the ICU stay, which included the occurrence of AKI, the need for kidney replacement therapy (KRT), red blood cell transfusions, the use of vasoactive drugs, mechanical ventilation, and mortality.

AKI was identified based on sCr levels and oliguria criteria. Two independent investigators systematically reviewed electronic medical records and laboratory tests using standardized forms. AKI was defined according to Kidney Disease Improving Global Outcomes (KDIGO) criteria (increase in sCr ≥ 0.3 mg/dL within 48h or ≥1.5 times baseline within 7 days) [21–23]. Baseline creatinine was defined as the lowest value within 3 months pre-admission or, if unavailable, the first ICU value.

Following the Surviving Sepsis Campaign and heart failure guidelines, Sepsis and ADHF diagnoses were established through medical record review, including clinical, laboratory, and imaging data [17,18]. A third investigator resolved any diagnostic discrepancies through consensus.

The primary endpoint was the occurrence of acute kidney injury (AKI) during the ICU stay, defined according to KDIGO criteria. The secondary endpoints included: (1) need for kidney replacement therapy (KRT), (2) requirement for red blood

cell transfusion, (3) need for mechanical ventilation, (4) mortality at day 28, and (5) long-term renal outcome defined as progression to dialysis-dependent stage V chronic kidney disease (CKD-V) during a 7-year follow-up.

## Statistical analysis

Categorical data are described as counts and percentages, and continuous data are defined as means and standard deviations or medians with interquartile ranges in the case of normal or non-normal distribution, respectively. The chi-square test was used to compare categorical data. The Kolmogorov–Smirnov test was used to assess the distribution of continuous data. Afterward, the comparisons between the groups were evaluated using the Student's T test for data with a normal distribution or by Mann-Whitney as non-parametric data. Pearson or Spearman tests were used to analyze the correlation between two continuous variables, as appropriate.

Binary logistic regression was performed for admission data related to outcomes in the ICU, using AKI as an eligible response variable after the principal comparative analysis. We eventually focused on and compared two new subgroups, divided based on the median Hb/RDW ratio values of all acutely ill patients. In our multivariate logistic regression model, we adjusted for several clinical variables to account for potential confounding effects. We incorporated inflammatory markers into our analysis, emphasizing C-reactive protein (CRP) levels measured at ICU admission. We selected CRP as the primary inflammatory marker because it is an acute-phase reactant that rises rapidly during inflammation and is routinely assessed in critically ill patients. Furthermore, previous studies have associated CRP with both RDW alterations and AKI development in various clinical settings [4,11].

Preexisting conditions were systematically considered in our analytical approach. Comorbidities such as diabetes mellitus, hypertension, and chronic kidney disease were included as categorical variables in our initial models, based on their established associations with AKI risk and potential influence on hematologic parameters. The SAPS 3 prognostic index retained in our final model, incorporates elements reflecting comorbidities and physiological disturbances, thereby capturing much of the clinical variability of preexisting conditions. We developed our final multivariate model using a backward stepwise selection process, beginning with all variables showing significant associations ($p < 0.10$) in univariate analyses: serum urea, SAPS 3, Hb/RDW ratio, and CRP levels. We evaluated the model for appropriate statistical fit and assessed potential variable interactions. The final model selection was based on statistical significance, clinical relevance, and overall performance in predicting AKI. Given that 117 patients developed AKI, the multivariable logistic regression respected the recommended rule of at least 10 outcome events per candidate predictor variable, thereby ensuring an adequate events-per-variable ratio for the final model. The $p < 0.05$ was considered significant. We used SPSS-version 22 (IBM, Armonk, New York, USA) and Microsoft Excel 365® (Microsoft, Redmond, Washington, USA) for statistical analyses.

## Compliance with ethical standards

The current study was conducted using the relevant ethical standards authorized by the local ethics committee of the Hospital Israelita Albert Einstein, São Paulo, Brazil (CAAE:01520218.9.0000.0071; CAAE:98877118.5.0000.0071; and CAAE:02628918.7.0000.0071). Informed consent was waived due to the declaration of data protection and all rights of research participants. The researchers analyzed exclusively de-identified and anonymized data. The present study has been conducted according to the principles expressed in the Declaration of Helsinki.

## Results

### Baseline population characteristics

We screened 386 consecutive critically ill ICU admissions and identified 202 eligible participants for this study after applying predefined exclusion criteria. Among the 119 patients with sepsis, 68 (57.1%) developed sepsis-associated AKI (S-AKI) within seven days of ICU admission. Additionally, among the 83 patients with acute decompensated heart failure (ADHF), 49 (59.0%) developed type 1 cardiorenal syndrome (T1-CRS) during the first week of admission (Fig 1).

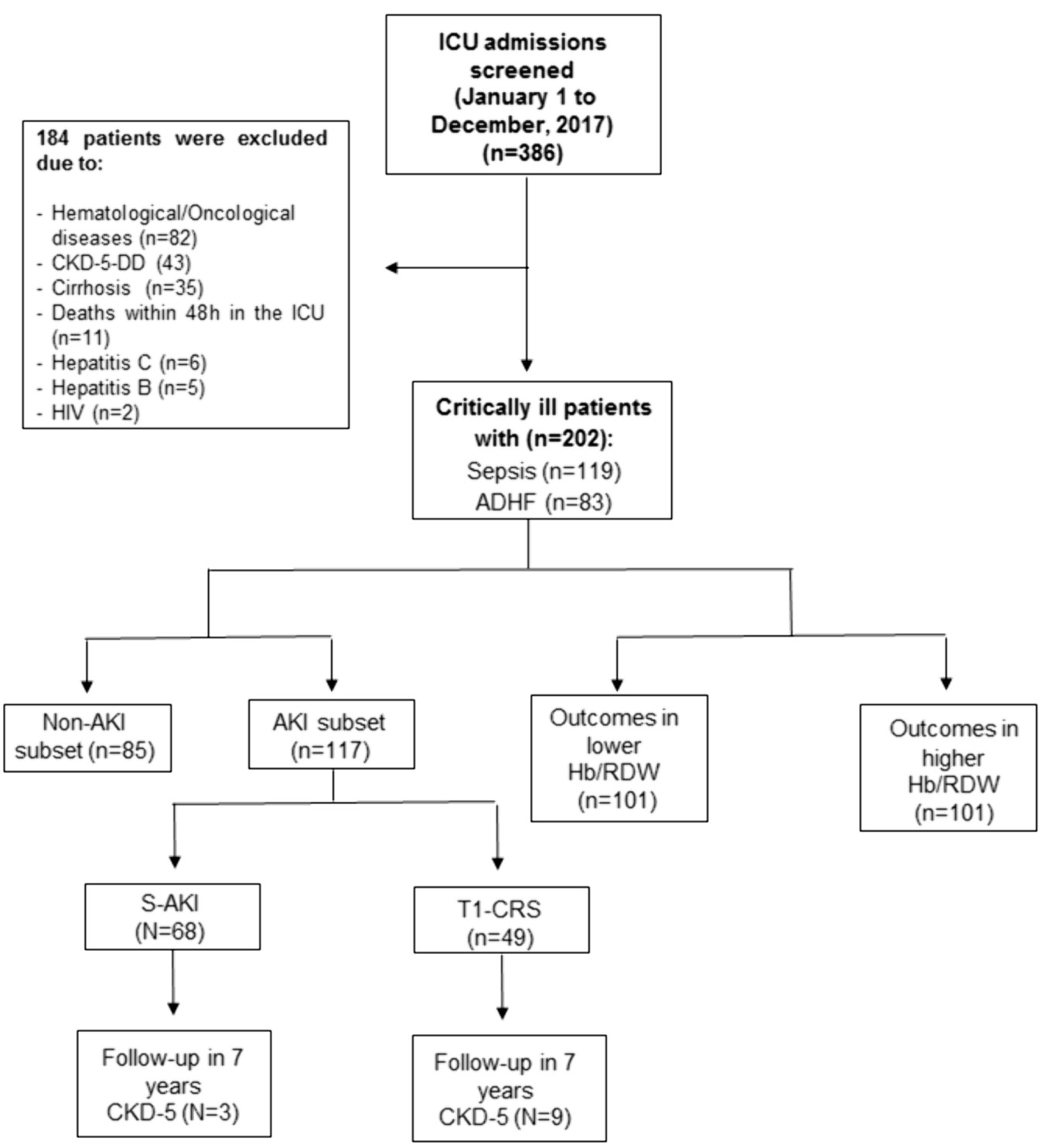

**Fig 1. Flow diagram of patient screening and inclusion process.** Flowchart depicting the screening of 386 consecutive ICU admissions diagnosed with sepsis or ADHF, application of exclusion criteria, and inclusion of 202 eligible patients. ADHF, acute decompensated heart failure; CKD-5-DD, chronic kidney disease stage 5 dialysis-dependent; ICU, intensive care unit; HIV, human immunodeficiency virus; AKI, acute kidney injury; Hb/RDW ratio, hemoglobin/red blood cell distribution width ratio; S-AKI, sepsis-associated acute kidney injury; T1-CRS, type 1 cardiorenal syndrome.

## Hb/RDW ratio and kidney function analysis

We conducted bivariate correlations using ICU admission data from 202 patients. Our analysis revealed several significant positive correlations. We observed a positive correlation between sCr and age (r = 0.14; p = 0.04), sCr and the SAPS3 prognostic index (r = 0.21; p = 0.002), serum urea and age (r = 0.31; p < 0.001), serum urea and RDW (r = 0.22; p = 0.002), and serum urea and the SAPS3 prognostic index (r = 0.26; p < 0.001).

Conversely, we observed negative correlations between sCr and Hb concentration (r = −0.31; p < 0.001), sCr and the Hb/RDW ratio (r = −0.30; p < 0.001), serum urea and $PaO_2/FiO_2$ (r = −0.15; p = 0.03), serum urea and Hb concentration (r = −0.21; p = 0.003), and serum urea and the Hb/RDW ratio (r = −0.27; p < 0.001), Hb/RDW ratio and age (r = −0.20; p = 0.005), and between Hb/RDW ratio and the SAPS3 prognostic index (r = −0.25; p < 0.001).

Among the 117 patients with AKI, 68 (58.1%) were identified as having S-AKI and 49 (41.9%) had T1-CRS (Fig 1). Additionally, the AKI severity was distributed as follows: 52 patients (44.4%) in KDIGO stage 1, 37 patients (31.6%) in KDIGO stage 2, and 28 patients (24%) in KDIGO stage 3.

Patients in the AKI group exhibited higher levels of sCr, serum urea, C-reactive protein, NT-proBNP, RDW, and SAPS 3 prognostic index. In contrast, patients in the non-AKI group had significantly higher Hb concentration and Hb/RDW ratio (Table 1).

In our comparative analysis between groups, patients with AKI showed significantly higher serum creatinine levels (1.61 ± 0.3 vs 1.22 ± 0.2 mg/dL; p < 0.001) and serum urea levels (86.3 ± 37.6 vs 61.9 ± 36.1 mg/dL; p < 0.001). Hemoglobin concentration was significantly lower in the AKI group (11.4 ± 1.9 vs 12.5 ± 1.7 g/dL; p < 0.001), while RDW was higher (15.7 ± 2.2 vs 14.9 ± 1.8%; p = 0.01). Notably, the Hb/RDW ratio was substantially lower in the AKI group compared to the non-AKI group (75.1 ± 1.6 vs 85.5 ± 1.9 g/dL; p < 0.001).

Serum urea, SAPS 3, and the Hb/RDW ratio at ICU admission were independently associated with AKI in patients with sepsis and ADHF. This binary logistic regression model also incorporated serum C-reactive protein levels (Table 2).

In our multivariate analysis, each 1 mg/dL increase in serum urea levels was associated with a 1.6% increase in AKI risk (OR 1.016; 95% CI 1.005–1.027; p = 0.003). Similarly, for each one-point increase in the SAPS 3 score, we observed a 4.0% increase in AKI risk (OR 1.040; 95% CI 1.008–1.074; p = 0.01). In contrast, each one-unit increase in the Hb/RDW ratio was associated with a 2.3% reduction in the risk of developing AKI (OR 0.977; 95% CI 0.959–0.996; p = 0.02), indicating a protective effect of a higher Hb/RDW ratio.

The final multivariable model retained serum urea, SAPS 3, and Hb/RDW ratio as independent predictors of AKI, while CRP was removed during stepwise selection. The model demonstrated satisfactory discrimination (AUROC = 0.763) and good calibration (Hosmer–Lemeshow $\chi^2$ = 8.283; p = 0.406), indicating appropriate overall fit and predictive performance.

In an alternative model using sCr instead of serum urea, the Hb/RDW ratio was not associated with AKI (odds ratio [OR] 0.986; 95% confidence interval [CI] 0.967–1.005). In contrast, sCr demonstrated a strong independent association with AKI (OR 195.940; 95% CI 31.105–1234.273).

On the seventh day of hospitalization, the levels of sCr were higher in the group of patients with AKI, measuring 2.72 ± 1.1 mg/dL and 1.04 ± 0.13 mg/dL (p < 0.001). Additionally, urea levels were significantly elevated at 104 ± 51 mg/dl and 61.8 ± 24.7 mg/dl (p < 0.001).

In our analysis of 202 acutely ill patients, we compared those who required KRT (n = 25) with those who did not during their 28-day stay in the ICU. The Hb/RDW ratio at ICU admission was lower in patients who developed AKI requiring KRT (Fig 2).

Patients who developed AKI requiring KRT (n = 25) had a significantly lower Hb/RDW ratio at ICU admission compared to those who did not require KRT (68.1 ± 16.5 vs 80.2 ± 18.4 g/dL; p = 0.002; mean difference −12.1 g/dl; 95% CI −19.1 to −5.0).

## Clinical outcomes

In the final stage of our comprehensive analysis, we compared the frequencies of clinical outcomes during the 28 days of ICU admission (Table 3). To facilitate this comparison, we divided the Hb/RDW ratio values by the median. It allowed

**Table 1. Comparison of demographic and clinical data on admission to the Intensive Care Unit between patients who developed acute kidney injury and those without acute kidney injury during hospitalization.**

| | AKI<br>n = 117 | Non-AKI<br>n = 85 | p-value |
|---|---|---|---|
| Age (Years) | 79.9 ± 12.6 | 77.7 ± 16.4 | 0.27 |
| Male (%) | 79 [67.5] | 67 [78.8] | 0.76 |
| Comorbidities, n [%] | | | |
| *Diabetes mellitus* | 47 [40.2] | 32 [37.6] | 0.72 |
| *Hypertension* | 65 [55.5] | 43 [50.6] | 0.48 |
| *CKD* | 15 [17.6] | 22 [18.8] | 0.83 |
| *Smoker* | 24 [20.5] | 22 [25.8] | 0.37 |
| MAP (mmHg) | 81.4 ± 21.1 | 76.8 ± 19.7 | 0.12 |
| Creatinine (mg/dL) | 1.61 ± 0.3 | 1.22 ± 0.2 | <0.001 |
| Urea (mg/dL) | 86.3 ± 37.6 | 61.9 ± 36.1 | <0.001 |
| Sodium (mEq/l) | 135 ± 14 | 135 ± 3.8 | 0.89 |
| Potassium (mEq/l) | 4.3 ± 0.8 | 4.2 ± 0.6 | 0.78 |
| CRP (mg/L) | 130 ± 28 | 98.9 ± 30.7 | 0.03 |
| Glycemia (mg/dL) | 142 ± 64 | 146 ± 59 | 0.65 |
| Total bilirubin (mg/dL) | 1.16 ± 0.8 | 0.9 ± 0.3 | 0.07 |
| NT-proBNP (pg/mL) | 1829 ± 458 | 1238 ± 730 | 0.30 |
| Hb (g/dL) | 11.4 ± 1.9 | 12.5 ± 1.7 | <0.001 |
| MCV (fl) | 91.7 ± 9.6 | 91.3 ± 9.7 | 0.87 |
| RDW (%)<br>Hb/RDW (g/dL) | 15.7 ± 2.2<br>75.1 ± 1.6 | 14.9 ± 1.8<br>85.5 ± 1.9 | 0.01<br><0.001 |
| WBC ($10^3$/µl) | 12.9 ± 2.1 | 11.2 ± 6.7 | 0.22 |
| Platelets ($10^3$/µl) | 193 ± 8.6 | 210 ± 13.7 | 0.27 |
| pH | 7.41 ± 0.07 | 7.42 ± 0.08 | 0.48 |
| $PaO_2$ (mmHg) | 102 ± 38.6 | 97.5 ± 37.6 | 0.25 |
| $PaCO_2$ (mmHg) | 31.9 ± 9.4 | 33.3 ± 8.5 | 0.28 |
| $HCO_3^-$ (mEq/L) | 20.2 ± 5.7 | 21.9 ± 5.3 | 0.07 |
| Lactate (mg/dL) | 20.6 ± 1.4 | 21.5 ± 2.3 | 0.72 |
| $PaO_2/FiO_2$ (mmHg) | 316 ± 11 | 337 ± 14 | 0.24 |
| SAPS3 | 74.1 ± 2.2 | 62.8 ± 1.9 | <0.001 |

*ADHF, acute decompensated heart failure; AKI, acute kidney injury; CKD, chronic kidney disease; MAP, mean arterial pressure; CRP, serum C Reactive Protein; NT-proBNP, N-terminal pro-B-type natriuretic peptide; Hb, hemoglobin concentration; MCV, mean corpuscular volume; RDW, red blood cell distribution width; WBC, white blood cell; $PaCO_2$, carbon dioxide; $PaO_2/FiO_2$, arterial oxygen pressure/inspired oxygen fraction; SAPS3, Simplified Acute Physiologic Score 3 prognostic index.*

us to categorize the results into two groups: those with lower Hb/RDW ratios (<79.93 g/dL) and those with higher ratios (>79.93 g/dL). Our findings indicated a higher frequency of AKI, oliguria, the need for KRT, mechanical ventilation, and RBC transfusion, as well as increased mortality rates in the group with a lower Hb/RDW ratio at the time of ICU admission.

Among the 202 critically ill patients, those with lower Hb/RDW ratio (<79.93 g/dL) exhibited significantly higher frequencies of AKI (72.3% vs 43.6%; p < 0.001; relative risk [RR] 1.66; 95% CI 1.29–2.14), need for KRT (18.8% vs 5.9%; p = 0.001; RR 3.17; 95% CI 1.32–7.60), mechanical ventilation (35.6% vs 21.8%; p = 0.01; RR 1.64; 95% CI 1.04–2.57),

**Table 2. Binary logistic regression of acute kidney injury as the response variable and their predictors on admission to Intensive Care Unit.**

| AKI vs non-AKI | OR | 95% CI for OR | | p-value |
|---|---|---|---|---|
| | | Lower | Upper | |
| Urea (mg/dL) | 1.016 | 1.005 | 1.027 | 0.003 |
| SAPS3 | 1.040 | 1.008 | 1.074 | 0.01 |
| Hb/RDW (g/dL) | 0.977 | 0.959 | 0.996 | 0.02 |
| CRP (mg/L) | 1.003 | 0.999 | 1.006 | 0.12 |

$R^2 = 0.745$; Global model (p = 0.02); OR, odds ratio; 95% CI, 95% confidence interval; AKI, acute kidney injury; CRP, serum C Reactive Protein; Hb/RDW, hemoglobin concentration/ red blood cell distribution width ratio; SAPS3, Simplified Acute Physiologic Score 3 prognostic index.

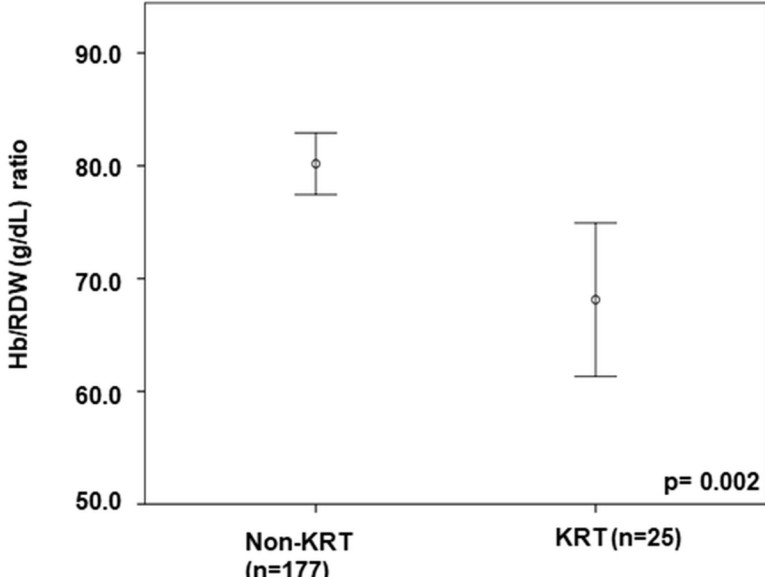

**Fig 2. A comparison of the Hb/RDW ratio at intensive care unit admission for patients who developed acute kidney injury requiring kidney replacement therapy during hospitalization.** Hb/RDW ratio, hemoglobin to red cell distribution width ratio; KRT, kidney replacement therapy.

**Table 3. Comparison of frequencies for clinical issues and outcomes within 28 days of intensive care unit hospitalization according to high and low Hb/RDW ratio group.**

| | Lower Hb/RDW n = 101 | Higher Hb/RDW n = 101 | p-value |
|---|---|---|---|
| AKI [%] | 73 [72.3] | 44 [43.5] | <0.001 |
| Oliguria [%] | 11 [10.9] | 4 [3.9] | 0.06 |
| KRT [%] | 20 [19.8] | 5 [4.9] | 0.001 |
| MV [%] | 37 [36.6] | 21 [20.8] | 0.01 |
| RBC transfusion [%] | 19 [18.8] | 1 [0.99] | <0.001 |
| Infection [%] | 65 [64.3] | 76 [75.2] | 0.09 |
| Mortality [%] | 26 [25.7] | 13 [12.9] | 0.02 |

AKI, acute kidney injury; Lower Hb/RDW ratio < 79.93 g/dL; Higher Hb/RDW ratio > 79.93 g/dL; Oliguria, urinary output < 400 mL/24 h; KRT, kidney replacement therapy; MV, mechanical ventilation requirement; RBC transfusion, red blood cell transfusion requirement.

red blood cell transfusion (18.8% vs 1.0%; p < 0.001; RR 19.00; 95% CI 2.59–139.26), and mortality (25.7% vs 12.9%; p = 0.02; RR 2.00; 95% CI 1.09–3.67) during the 28 days of ICU stay.

Eventually, we compared the ICU admission data of a subset of patients who developed S-AKI (n = 68) with another group, T1-CRS (n = 49), within one week of admission. Our analysis revealed that the T1-CRS subset had a higher frequency of chronic kidney disease and RDW levels. In contrast, the S-AKI subset exhibited higher Hb concentrations, a higher Hb/RDW ratio, increased arterial lactate levels, and higher ejection fraction.

During the 7-year follow-up, 9 patients (10.8%) in the T1-CRS group and 3 patients (2.5%) in the sepsis group progressed to dialysis-dependent stage V chronic kidney disease (CKD-V), with median progression times of 3 years (range 1–5) and 4 years (range 3–5), respectively (p = 0.031). Overall, dialysis dependence typically developed between 1 and 5 years after the initial critical illness episode (S1 Table).

## Discussion

This proof-of-concept study demonstrates that the Hb/RDW ratio, an easily accessible laboratory index, is associated with both acute and long-term kidney outcomes in critically ill patients. The Hb/RDW ratio at ICU admission might help predict the occurrence of AKI in acutely ill patients with sepsis or ADHF. Additionally, patients with a lower Hb/RDW ratio at ICU admission exhibited a higher frequency of worse clinical outcomes, such as the need for KRT, RBC transfusions, mechanical ventilation, and mortality within 28 days of hospitalization.

Anemia is common among critically ill patients due to several factors, including increased inflammation, blood loss from trauma or surgical procedures, and relative or functional erythropoietin deficiency [1,24–26]. The literature showed that perioperative anemia is linked to AKI, cardiovascular disease, infection, and a higher 30-day mortality rate [27]. Moreover, a study created a direct prediction tool named the Simple Postoperative AKI Risk, which includes anemia in its assessment of AKI for patients undergoing noncardiac surgery [28].

We aimed to investigate the relationship between the Hb/RDW ratio and AKI in critically ill patients suffering from sepsis and ADHF because these conditions are common causes of AKI in the ICU and are often linked to anemia, inflammation, and worse outcomes [4,13–15,29,30]. The RDW measures the variability in the size of circulating erythrocytes. It is a valuable tool for diagnosing anemia and can be obtained from a complete blood count [15]. Moreover, RDW has been associated with inflammation [31]. As a result, RDW is increasingly recognized as an inflammatory marker and a prognostic biomarker for cardiovascular and kidney diseases [4,32,33].

In the current study, we observed a high incidence of AKI and anemia in patients with sepsis or ADHF. We confirmed a strong relationship between AKI and anemia in these critically ill patients. Our findings indicated a significant prevalence of both conditions among individuals with sepsis and ADHF. We demonstrated a compelling link between Hb/RDW ratio, anemia, and AKI in critically ill individuals. Specifically, our results showed an inverse relationship between kidney function markers and anemia indicators. We found a negative correlation between sCr and Hb/RDW ratio and serum urea and Hb concentration. It is all-important to be able to perform screening and stratification of clinical phenotypes in acutely ill patients with sepsis and ADHF who are decompensated, as they may be at risk for developing AKI and experiencing worse clinical outcomes.

Additionally, the inverse relationship between the Hb/RDW ratio and the SAPS 3 prognostic index underscores underscores that patients with a lower Hb/RDW ratio at the time of ICU admission have greater disease severity and increased risk of AKI development. Also, the group of patients who experienced AKI during ICU stay showed higher levels of RDW and a lower Hb/RDW ratio. Furthermore, in our study, patients with lower Hb/RDW values at ICU admission experienced worse outcomes during 28 days of hospitalization, including the need for RBC transfusion, mechanical ventilation, KRT, non-survival, and AKI. These clinical parameters indicate greater severity and worse prognosis in patients with T1-CRS and S-AKI.

In our multivariate analysis, we found that each 1 g/dL increase in the Hb/RDW ratio greater than 79.93 g/dL was independently associated with a significant 2.3% reduction in the risk of AKI among critically ill patients suffering from sepsis

and ADHF when using urea as a kidney function marker. Using sCr as the marker, the Hb/RDW ratio did not show an independent association for AKI. It may be because our study population defined AKI based on sCr levels.

Our group has previously researched and published findings on the association between AKI, inflammation, and anemia [1]. We have also investigated the connection between RDW and AKI associated with sepsis [4]. Numerous studies have emphasized the association between RDW levels and outcomes in AKI, including mortality in patients undergoing continuous kidney replacement therapy [34], as well as overall mortality rates [33,34]. Additionally, a study reported a link between a lower Hb/RDW ratio and AKI caused by iodinated contrast in patients undergoing coronary angiography [35]. Given these findings, there is an urgent need for further research on the Hb/RDW ratio to determine if it could serve as a potentially inexpensive and readily available predictive marker for AKI, particularly in cases of sepsis and ADHF, which are common among hospitalized patients. Therefore, we aimed to explore the relationship between the Hb/RDW ratio and AKI during sepsis and ADHF. Understanding this connection could enhance outcomes for critically ill AKI patients [26,36]. Likewise, RDW levels were higher in patients with T1-CRS. On the other hand, the Hb/RDW ratio was lower in T1-CRS patients than in those with S-AKI (S1 Table).

In the exploration of acute kidney injury (AKI) predictors, our findings demonstrate that the hemoglobin to red blood cell distribution width (Hb/RDW) ratio serves as a valuable prognostic tool for AKI that complements established biomarkers. Traditional AKI indicators include serum creatinine, urine output, while emerging biomarkers include neutrophil gelatinase-associated lipocalin (NGAL), kidney injury molecule-1 (KIM-1), and interleukin-18 (IL-18). These specialized biomarkers, while informative in specific clinical contexts, often require costly assays that may not be universally available [37]. In contrast, the Hb/RDW ratio utilizes parameters routinely obtained from complete blood counts, making it immediately accessible and particularly advantageous in resource-limited settings.

Previous studies have reinforced the significance of RDW as an isolated factor linked to AKI and mortality, with some demonstrating its independent predictive value for mortality among AKI patients undergoing continuous renal replacement therapy [33]. Our findings build on this foundation by suggesting that the Hb/RDW ratio may enhance prognostic capability over the individual parameters alone.

Recent studies have highlighted the potential of combined biomarkers in predicting AKI. Researchers have proposed nomograms incorporating clinical parameters and laboratory data to enhance the prediction of sepsis-associated AKI, while others have evaluated the prognostic value of combining RDW with the neutrophil-to-lymphocyte ratio [11,26]. Aligning with this evolving framework, our study supports the Hb/RDW ratio as a practical risk stratification tool that reflects the interplay between anemia and anisocytosis and suggests broader systemic involvement in critically ill patients.

Furthermore, our logistic regression model demonstrated that the Hb/RDW ratio retains predictive value alongside SAPS 3 and serum urea levels. This supports the integration of the Hb/RDW ratio into existing risk stratification approaches and aligns with recent trends favoring composite markers for enhanced prognostication [38,39]. Thus, the Hb/RDW ratio may complement existing clinical scores and provide additional insight into systemic processes like inflammation, anemia, and organ dysfunction, helping to refine AKI risk stratification in diverse ICU populations [40].

Our findings indicate a significant association between a lower Hb/RDW ratio and increased AKI prevalence in critically ill patients with sepsis and ADHF. However, several potential confounding factors could influence this relationship.

Age-related factors significantly influence this association. We found a negative correlation between the Hb/RDW ratio and age ($r = -0.20$; $p = 0.005$), suggesting that older patients tend to have lower ratios due to inherent biological changes. With aging, patients typically exhibit increased baseline inflammation, reduced bone marrow reserve, and declining renal function, which affect both hematological parameters and vulnerability to AKI [32,41]. Furthermore, a significant number of our cohort had multiple comorbidities, such as diabetes mellitus (40.2% of AKI patients) and hypertension (55.5% of AKI patients), which are recognized as factors that can independently impact both kidney function and hematological status [32,42].

Inflammation plays a crucial role in critically ill patients. The inflammatory environment can impair erythropoiesis, increasing RDW values, while simultaneously contributing to AKI through mechanisms including microvascular dysfunction and oxidative stress [32,33]. Although we adjusted for C-reactive protein levels in our multivariate analysis, unmeasured inflammatory mediators might still affect both the Hb/RDW ratio and renal function [43,44]. Notably, patients with sepsis showed different Hb/RDW ratio patterns compared to those with ADHF, suggesting that underlying inflammatory states may influence this relationship [45,46].

Medication exposure represents another potential confounder not fully addressed in our analysis. Common treatments for critically ill patients—including antibiotics, diuretics, and vasoactive agents—may independently affect kidney function and hematological parameters, potentially confounding the relationship between Hb/RDW ratio and AKI [33,47]. Furthermore, hemodilution and fluid management strategies can alter both hemoglobin concentrations and apparent AKI severity, adding complexity to these interrelationships [48].

An alternative explanation for the findings may involve oxidative stress, which could represent a common pathway affecting both red blood cell parameters and kidney function. Increased levels of oxidative stress have been linked to heightened RDW values and are also critical in the pathogenesis of AKI, particularly among patients suffering from sepsis and ADHF [32,41]. Endothelial dysfunction, recurrent in critically ill populations, could further contribute to a simultaneous decline in hemoglobin levels, RDW alterations, and kidney injury [49,50].

Our finding that a lower Hb/RDW ratio at ICU admission is associated with the development of AKI and poorer clinical outcomes in patients with sepsis and ADHF is consistent with recent evidence highlighting RDW as an independent prognostic biomarker across various critical and cardiovascular conditions. In patients with heart failure, a recent study demonstrated that elevated RDW levels correlate with greater disease severity, reinforcing the role of this hematologic index as a marker of both ventricular dysfunction and systemic inflammation [51]. Similarly, a recent article reported that the RDW-to-albumin ratio on admission predicted 28-day all-cause mortality in patients with acute pancreatitis, supporting the concept that simple RDW-based indices can reflect systemic inflammation and physiological reserve -mechanisms also implicated in the pathophysiology of sepsis-associated AKI observed in our cohort [52].

Moreover, studies in non-critical populations have further underscored the systemic relevance of RDW as a cardiorenal risk marker. Elevated RDW levels were associated with all-cause and cardiovascular mortality in individuals with non-alcoholic fatty liver disease (NAFLD), suggesting that RDW reflects chronic inflammation and multisystem dysfunction [53]. In the context of acute respiratory distress syndrome (ARDS), it was observed that both absolute RDW levels and their dynamic changes during hospitalization were correlated with the incidence of AKI and 28-day mortality, demonstrating the value of RDW as an indicator of disease severity and clinical trajectory [54]. Complementarily, a longitudinal analysis of septic patients with AKI showed that RDW dynamics over time robustly predicted in-hospital mortality, aligning with our results and reinforcing that Hb/RDW at ICU admission is a clinically useful marker for risk stratification and outcome prediction in critically ill populations [55].

Despite rigorous statistical adjustments for numerous clinical variables in our multivariate analysis, residual confounding may persist due to the complex interplay of factors affecting critically ill patients. Since our study is observational, it provides valuable insight into associations but cannot establish causation between the Hb/RDW ratio and AKI. Future prospective research incorporating a broader range of confounding variables is essential to further clarify this complex relationship.

Our study shows some limitations. First, it is a single-center study. Second, the sample size may have influenced the results. Third, some medications used in critically ill patients, such as antibiotics and vasopressors, and previous use of medications that interfere with the renin-angiotensin system might influence the Hb/RDW ratio results and acute renal dysfunction. Despite these limitations, one of the main strengths of the current study is that it relies on routine blood tests conducted at the time of ICU admission, making it a low-cost investigation.

## Conclusion

A lower Hb/RDW ratio is strongly and independently associated with the occurrence of AKI in critically ill patients with sepsis and acute decompensated heart failure. This simple hematologic index may help identify high-risk clinical phenotypes early in the course of critical illness and could also reflect long-term renal vulnerability. Given the exploratory and proof-of-concept nature of this study, further prospective research in larger and more diverse cohorts is warranted to validate the predictive and prognostic utility of the Hb/RDW ratio in intensive care settings.

## Supporting information

**S1 Table. Comparison of demographic and clinical data between sepsis-associated acute kidney injury and type 1-cardiorenal syndrome patients from intensive care unit admission.** AKI, acute kidney injury; S-AKI, sepsis-associated acute kidney injury; T1-CRS, Cardiorenal Syndrome type 1; CKD, chronic kidney disease; CRP, C Reactive Protein; NT-proBNP, N-terminal pro-B-type natriuretic peptide; Hb, hemoglobin concentration; MAP, mean arterial pressure; MCV, mean corpuscular volume; RDW, red blood cell distribution width; WBC, white blood cell; $PaO_2/FiO_2$, arterial oxygen pressure/inspired oxygen fraction; SAPS3, Simplified Acute Physiologic Score 3 prognostic index.
(DOCX)

## Acknowledgments

We thank all the Einstein Hospital Israelita professionals and technicians involved in this study for their support. We also thank Professor Nestor Schor (in memoriam) for his assistance and consideration.

## Author contributions

**Conceptualization:** Petherson Mendonça dos Santos, Isabele Pardo, Beatriz Moreira Silva, Miguel Angelo Goes.

**Data curation:** Petherson Mendonça dos Santos, Isabele Pardo, Maria Carolina Borges Pereira de Almeida, Rafael Santana de Oliveira, Fernanda Gabas Miglioli, Beatriz Mota Busnardo, Felipe Prieto Siqueira, Daniela Harsanyi, Andreia Pardini, Bruno Bravim, Anna Carolina de Rizzo Cantoni Rosa, Kirliane de Sousa Rodrigues Lacerda, Yvve Priscila Gatto, Lucas Andrade Pinheiro, Rayane Alves Moreira, Mateus Américo Galvão Santos, João Paulo Fonseca da Silva, João Marcos Santos da Rocha, Remo Holanda M Furtado, Beatriz Moreira Silva, Miguel Angelo Goes.

**Formal analysis:** Isabele Pardo, Maria Carolina Borges Pereira de Almeida, Rafael Santana de Oliveira, Fernanda Gabas Miglioli, Beatriz Mota Busnardo, Felipe Prieto Siqueira, Daniela Harsanyi, Andreia Pardini, Bruno Bravim, Kirliane de Sousa Rodrigues Lacerda, Yvve Priscila Gatto, Lucas Andrade Pinheiro, Rayane Alves Moreira, João Paulo Fonseca da Silva, Remo Holanda M Furtado, Beatriz Moreira Silva, Miguel Angelo Goes.

**Investigation:** Petherson Mendonça dos Santos, Isabele Pardo, Maria Carolina Borges Pereira de Almeida, Rafael Santana de Oliveira, Fernanda Gabas Miglioli, Beatriz Mota Busnardo, Felipe Prieto Siqueira, Daniela Harsanyi, Andreia Pardini, Bruno Bravim, Anna Carolina de Rizzo Cantoni Rosa, Kirliane de Sousa Rodrigues Lacerda, Yvve Priscila Gatto, Lucas Andrade Pinheiro, Rayane Alves Moreira, Mateus Américo Galvão Santos, João Paulo Fonseca da Silva, João Marcos Santos da Rocha, Remo Holanda M Furtado, Beatriz Moreira Silva, Miguel Angelo Goes.

**Methodology:** Petherson Mendonça dos Santos, Isabele Pardo, Maria Carolina Borges Pereira de Almeida, Rafael Santana de Oliveira, Fernanda Gabas Miglioli, Beatriz Mota Busnardo, Felipe Prieto Siqueira, Daniela Harsanyi, Andreia Pardini, Bruno Bravim, Anna Carolina de Rizzo Cantoni Rosa, Kirliane de Sousa Rodrigues Lacerda, Yvve Priscila Gatto, Lucas Andrade Pinheiro, Rayane Alves Moreira, Mateus Américo Galvão Santos, João Paulo Fonseca da Silva, João Marcos Santos da Rocha, Remo Holanda M Furtado, Beatriz Moreira Silva, Miguel Angelo Goes.

**Project administration:** Petherson Mendonça dos Santos, Isabele Pardo, Rafael Santana de Oliveira, Fernanda Gabas Miglioli, Beatriz Mota Busnardo, Felipe Prieto Siqueira, Daniela Harsanyi, Andreia Pardini, Bruno Bravim, Kirliane de Sousa Rodrigues Lacerda, Yvve Priscila Gatto, Lucas Andrade Pinheiro, Rayane Alves Moreira, Mateus Américo Galvão Santos, Remo Holanda M Furtado, Miguel Angelo Goes.

**Resources:** Petherson Mendonça dos Santos, Rafael Santana de Oliveira, Beatriz Mota Busnardo, Daniela Harsanyi, Bruno Bravim, Anna Carolina de Rizzo Cantoni Rosa, Kirliane de Sousa Rodrigues Lacerda, Yvve Priscila Gatto, Lucas Andrade Pinheiro, Rayane Alves Moreira, Mateus Américo Galvão Santos, João Paulo Fonseca da Silva, João Marcos Santos da Rocha, Remo Holanda M Furtado, Miguel Angelo Goes.

**Software:** Petherson Mendonça dos Santos, Isabele Pardo, Fernanda Gabas Miglioli, Felipe Prieto Siqueira, Andreia Pardini, Bruno Bravim, Anna Carolina de Rizzo Cantoni Rosa, Kirliane de Sousa Rodrigues Lacerda, Yvve Priscila Gatto, Lucas Andrade Pinheiro, Mateus Américo Galvão Santos, João Paulo Fonseca da Silva, João Marcos Santos da Rocha, Beatriz Moreira Silva, Miguel Angelo Goes.

**Supervision:** Petherson Mendonça dos Santos, Isabele Pardo, Felipe Prieto Siqueira, Remo Holanda M Furtado, Beatriz Moreira Silva, Miguel Angelo Goes.

**Validation:** Petherson Mendonça dos Santos, Isabele Pardo, Maria Carolina Borges Pereira de Almeida, Rafael Santana de Oliveira, Felipe Prieto Siqueira, Daniela Harsanyi, Andreia Pardini, Bruno Bravim, Anna Carolina de Rizzo Cantoni Rosa, Kirliane de Sousa Rodrigues Lacerda, Yvve Priscila Gatto, Lucas Andrade Pinheiro, Rayane Alves Moreira, Mateus Américo Galvão Santos, João Paulo Fonseca da Silva, João Marcos Santos da Rocha, Remo Holanda M Furtado, Beatriz Moreira Silva, Miguel Angelo Goes.

**Visualization:** Petherson Mendonça dos Santos, Isabele Pardo, Rafael Santana de Oliveira, Fernanda Gabas Miglioli, Beatriz Mota Busnardo, Felipe Prieto Siqueira, Andreia Pardini, Anna Carolina de Rizzo Cantoni Rosa, Kirliane de Sousa Rodrigues Lacerda, Yvve Priscila Gatto, Lucas Andrade Pinheiro, Rayane Alves Moreira, Mateus Américo Galvão Santos, João Marcos Santos da Rocha, Remo Holanda M Furtado, Beatriz Moreira Silva.

**Writing – original draft:** Petherson Mendonça dos Santos, Isabele Pardo, Beatriz Mota Busnardo, Daniela Harsanyi, Yvve Priscila Gatto, Remo Holanda M Furtado, Beatriz Moreira Silva, Miguel Angelo Goes.

**Writing – review & editing:** Petherson Mendonça dos Santos, Isabele Pardo, Andreia Pardini, Rayane Alves Moreira, Remo Holanda M Furtado, Beatriz Moreira Silva, Miguel Angelo Goes.

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
