## [Editor Report · Decision Letter 0]

24 Mar 2025

Dear Dr. Mendonça dos Santos,

Thank you for submitting your manuscript to PLOS ONE. After careful consideration, we feel that it has merit but does not fully meet PLOS ONE’s publication criteria as it currently stands. Therefore, we invite you to submit a revised version of the manuscript that addresses the points raised during the review process.

We look forward to receiving your revised manuscript.

Kind regards,

Elabbass Abdelmahmuod

Academic Editor

PLOS ONE

3. We notice that your supplementary tables are included in the manuscript file. Please remove them and upload them with the file type 'Supporting Information'. Please ensure that each Supporting Information file has a legend listed in the manuscript after the references list.

Additional Editor Comments:

Expand the discussion on potential confounders and alternative explanations for the observed associations.

Clarify adjustments made for inflammatory markers and preexisting conditions.

Provide confidence intervals for key statistical findings in the results section.

Discuss how findings compare to existing literature on AKI predictors.

Revise minor language issues to improve readability and clarity.

---

## [Author Response · Author response to Decision Letter 1]

2 May 2025

1- Expand the discussion on potential confounders and alternative explanations for the observed associations.

We thank the editor for this important suggestion. Our discussion section has significantly expanded to address potential confounders and alternative explanations for the observed associations (pages 18-19). Specifically, we have added comprehensive paragraphs addressing age-related factors and their impact on both Hb/RDW ratio and AKI risk, the role of inflammation as a potential confounder, medication exposure effects on kidney function and hematological parameters, the impact of comorbidities on study outcomes, also oxidative stress as a standard pathway affecting both parameters and alternative explanations involving endothelial dysfunction.

We have included appropriate references to support these discussions (references 41-50).

2 - Clarify adjustments made for inflammatory markers and preexisting conditions.

In the Methods section, we have clarified our methodological approach for adjusting inflammatory markers and preexisting conditions (page 08). Now, we explicitly describe how C-reactive protein (CRP) was selected as the primary inflammatory marker, the rationale for CRP selection and its measurement timing, how preexisting conditions (diabetes mellitus, hypertension, chronic kidney disease) were systematically incorporated, the role of SAPS 3 prognostic index in capturing clinical variability, and our backward stepwise selection process for the multivariate model.

3 - Provide confidence intervals for key statistical findings in the results section.

As requested, we have added confidence intervals throughout the Results section for all key statistical findings. It includes comparative analyses between groups with confidence intervals for differences, odds ratios with 95% confidence intervals from multivariate analysis, relative risks with 95% confidence intervals for clinical outcomes, and mean differences with 95% confidence intervals for continuous variables.

All additions are marked in the revised manuscript, pages 12 to 15.

4 - Discuss how findings compare to existing literature on AKI predictors

We have significantly expanded our discussion to compare our findings with existing literature on AKI predictors (pages 17-18). We added a comprehensive section that compares the Hb/RDW ratio with traditional biomarkers and discusses their respective advantages and limitations. We particularly emphasize the practical benefits of using the Hb/RDW ratio, especially its potential in resource-limited settings. This section also contextualizes our findings within previous research on RDW as a predictor of clinical outcomes, discusses the emerging trend of using combined biomarkers for improved risk stratification, and addresses the specific implications for heart failure patients and cardio-renal interactions.

Additionally, we have added a new paragraph that further strengthens this comparison by highlighting how the Hb/RDW ratio retains its predictive value alongside established clinical parameters (SAPS 3 and serum urea levels) in our logistic regression model. This new section not only clarifies how the Hb/RDW ratio can be integrated into current risk stratification strategies, but also reinforces its practical relevance, aligning with emerging trends that favor the use of composite markers to enhance prognostication in critical care settings. We have ensured that all new additions are supported by up-to-date and relevant references. These citations were carefully selected to better contextualize our findings within the evolving landscape of AKI prediction research. We hope this strengthens the overall clarity and credibility of our results.

5 - Revise minor language issues to improve readability and clarity.

We thank the editor for the valuable suggestion to improve the clarity and readability of our manuscript. In response, we carefully revised the text throughout, focusing on sentence structure, terminology, and overall flow. Specifically, we restructured complex sentences in the Abstract (page 4), Methods (pages 6–7), and Results (pages 9–15). We also standardized terminology and improved consistency in reporting statistical results, particularly in Tables 1–3 and their corresponding descriptions. The clarity of correlations and statistical findings in the Results section (pages 11–15) was enhanced, and we refined the description of our analytical approach in the Methods (page 8). Additionally, we improved sentence structure and paragraph organization in the Discussion (pages 15–19) to better convey our findings and their implications. All changes were made without altering the scientific content or conclusions. We believe these revisions substantially improve the overall readability and presentation of the manuscript.

Additional Journal Requirements:

1. Manuscript formatting

Response: We confirm that the manuscript has been formatted in full accordance with the PLOS ONE guidelines, using the provided templates. All sections were reviewed to ensure compliance with formatting requirements, including margins, line spacing, heading structure, and reference style.

2. Data availability statement

Response: We have updated our Data Availability Statement to comply with PLOS ONE’s open data policy. The data supporting the findings of this study are available on Figshare at the following DOI: 10.6084/m9.figshare.28200248. The dataset includes only de-identified patient information and is fully compliant with data protection regulations and the protocol approved by our local ethics committee. Currently, the dataset is accessible via the DOI link with limited visibility and will be made fully public upon formal publication, in accordance with Figshare’s embargo feature.

3. Supplementary tables

Response: As requested, we have removed the supplementary tables from the main manuscript and uploaded them separately as Supporting Information files. Legends for each file have been added after the references section in the manuscript.

4. Reference list review

Response: We have thoroughly reviewed our reference list to ensure it is complete and accurate. All references cited in the manuscript are included in the reference list, and vice versa. We have confirmed that no retracted papers are cited. Additionally, we have added several new references (38-50) to support the expanded discussion on AKI predictors and potential confounders.

These revisions have significantly strengthened the manuscript, and we believe we have addressed all of the editor's concerns. We hope that the revised manuscript now meets the standards for publication in PLOS ONE.

Thank you for your time and consideration. We look forward to your feedback.

---

## [Decision Letter · Decision Letter 1]

29 Sep 2025

Dear Dr. Mendonça dos Santos,

Thank you for submitting your manuscript to PLOS ONE. After careful consideration, we feel that it has merit but does not fully meet PLOS ONE’s publication criteria as it currently stands. Therefore, we invite you to submit a revised version of the manuscript that addresses the points raised during the review process.

We look forward to receiving your revised manuscript.

Kind regards,

Helen Howard

Staff Editor

PLOS ONE

Journal Requirements:

Additional Editor Comments:

- The Editorial team has assessed your manuscript and concerns were raised over the small sample size and the fact that the data are from 2017. In order for us to consider the manuscript further, as part of this revision, the sample size will need to be increased and more recent data will need to be included. Please note that a failure to increase the sample size and/or include more recent data may lead to the rejection of the revised manuscript.

Reviewers' comments:

Reviewer's Responses to Questions

**Comments to the Author**

Reviewer #1: (No Response)

2. Is the manuscript technically sound, and do the data support the conclusions?

Reviewer #1: Yes

3. Has the statistical analysis been performed appropriately and rigorously?

Reviewer #1: Yes

4. Have the authors made all data underlying the findings in their manuscript fully available?

Reviewer #1: Yes

5. Is the manuscript presented in an intelligible fashion and written in standard English?

Reviewer #1: Yes

Reviewer #1: Interesting paper. Some issues should be added.

1) I do not understnt selection of patients. Actually in a relevant hospital in 1 year few patients seems to have been enrolled, being potentially not consecutive and leading to selection bias. please comment

2) primary and secondary end points shiuld be clearly stated

3) sample size calculation should be added

4)it is not clear how variables were included in the multivarite analysis In particulara lto of variables (3on 4) were significant potentlly leading to low accuracy of the model

**Do you want your identity to be public for this peer review?** For information about this choice, including consent withdrawal, please see our Privacy Policy

Reviewer #1: **Yes:** Fabrizio D'Ascenzo

---

## [Author Response · Author response to Decision Letter 2]

10 Nov 2025

We have carefully revised our manuscript, "Association between Hemoglobin/Red Blood Cell Distribution Width Ratio and Acute Kidney Injury in Sepsis and Heart Failure Patients" (PONE-D-25-12369).

We are grateful for the time and effort that both the editorial team and the reviewers have dedicated to evaluating our manuscript. We are pleased to hear that the reviewers have recognized the merit of our work, and we appreciate their feedback.

Following your advice, we have addressed all the points raised during the review process and added new information to the manuscript. Below, we provide a detailed point-by-point response to each comment:

We appreciate the editor’s concern regarding the 2017 dataset. Our study included all consecutive ICU admissions for sepsis or acute decompensated heart failure (ADHF) during that year, with complete laboratory and outcome data retrieved from a unified institutional database before changes in the hospital’s electronic record system. A considerable number of ICU deaths occurred within the first 48 hours of admission, making it impossible to evaluate kidney injury dynamics in these cases and justifying their exclusion from the analysis.

To strengthen the dataset and respond to the editor’s request for more recent information, we conducted a long-term follow-up extending to July 2024, assessing renal outcomes such as progression to dialysis-dependent stage V chronic kidney disease (CKD-V). This longitudinal extension expanded the temporal scope of the study, allowing us to examine both short-term (AKI) and long-term (CKD progression) kidney outcomes (page 6).

We were unable to include additional patient recruitment beyond 2017 because of ethical restrictions and structural changes in the hospital’s electronic data system. In addition, the SARS-CoV-2 pandemic profoundly altered ICU admission patterns, making it difficult to merge pre- and post-pandemic data in a methodologically consistent way. For these reasons, the dataset we present represents a complete and homogeneous annual cohort, now supported by validated longitudinal follow-up. We believe this design minimizes selection bias and provides clinically relevant insight into both acute and chronic renal outcomes. Finally, the prognostic question explored—the predictive value of the Hb/RDW ratio in critically ill patients—remains highly relevant today, as confirmed by several studies published between 2024 and 2025.

Reviewer Comments

1 - I do not understand patient selection. Actually in a relevant hospital in 1 year few patients seems to have been enrolled, being potentially not consecutive and leading to selection bias

We thank the reviewer for this important observation. We have clarified the screening and enrollment process in the revised Methods section. This study included all consecutive ICU admissions diagnosed with sepsis or acute decompensated heart failure (ADHF) during a fixed one-year period (January 1 to December 31, 2017). A total of 386 critically ill adults were initially screened. After applying strict predefined exclusion criteria (hematologic/oncologic diseases, CKD-5 on dialysis, cirrhosis, chronic viral infections, and death within the first 48 hours in the ICU), 202 patients were deemed eligible and included in the final cohort. This approach ensures a complete consecutive annual cohort, thereby minimizing selection bias. To further enhance transparency, Figure 1 has been updated to clearly display the flow of patient screening, exclusion, and inclusion with exact numbers and the explicit note that admissions were consecutive (page 10).

2 - Primary and secondary end points should be clearly stated.

We have clearly defined the study endpoints in the revised Methods section (pages 6-8). The primary endpoint was the occurrence of acute kidney injury (AKI) during the ICU stay, according to KDIGO criteria; AKI was selected as the primary outcome given its central role in the pathophysiology of our population (patients with sepsis or acute decompensated heart failure). The secondary endpoints included: (1) need for kidney replacement therapy, (2) requirement for red blood cell transfusion, (3) need for mechanical ventilation, and (4) mortality at day 28 and (5) long-term renal outcome defined as progression to dialysis-dependent stage V chronic kidney disease (CKD-V) during a 7-year follow-up.

3 - Sample size calculation should be added.

We appreciate the reviewer’s suggestion. As a proof-of-concept, hypothesis-generating observational study based on a complete annual cohort of all consecutive eligible ICU admissions, a traditional a priori sample size calculation was not applicable to our design. Instead, to evaluate the adequacy of our sample for multivariable analysis, we adhered to the established methodological principle of at least 10 outcome events per candidate predictor variable. With 117 AKI events, our dataset would support up to 11 candidate predictors; in practice, only four variables were entered into the initial model and three were retained in the final model. This yielded an events-per-variable ratio well above recommended thresholds, ensuring statistical robustness and minimizing the risk of overfitting. Accordingly, the sample size is appropriate for the analytical approach adopted in this exploratory context.

All additions are marked in the revised manuscript, page 8.

4 - It is not clear how variables were included in the multivariate analysis. In particular, a lot of variables (3 on 4) were significant potentially leading to low accuracy of the model.

We thank the reviewer for this insightful comment. We have now clarified the variable selection process in the revised Statistical Analysis section. All variables showing an association with AKI at p < 0.10 in univariate analyses (serum urea at admission, SAPS 3, Hb/RDW ratio, and C-reactive protein [CRP]) were entered into a multivariable logistic regression model using a backward stepwise elimination approach (removal threshold p ≥ 0.10). This process was conducted while adhering to the methodological principle of at least 10 outcome events per candidate predictor (117 AKI events in total). The final parsimonious model retained serum urea, SAPS 3, and the Hb/RDW ratio, with CRP being removed during the stepwise selection process. The model demonstrated satisfactory discrimination (AUROC = 0.763) and adequate calibration (Hosmer–Lemeshow χ² = 8.283; p = 0.406). These revisions clarify the criteria for variable inclusion, demonstrate that overfitting was minimized, and confirm that the resulting model achieved appropriate accuracy and internal validity.All corresponding additions and clarifications are marked in the revised manuscript (page 13)

These revisions have significantly strengthened the manuscript, and we believe we have addressed all of the editor's concerns. We hope that the revised manuscript now meets the standards for publication in PLOS ONE.

Thank you for your time and consideration. We look forward to your feedback.

Sincerely,

On behalf of all co-authors,

Petherson Mendonça dos Santos

Division of Nephrology, Federal University of São Paulo, Brazil

---

## [Decision Letter · Decision Letter 2]

8 Feb 2026

Association between Hemoglobin/Red Blood Cell Distribution Width Ratio and Acute Kidney Injury in Sepsis and Heart Failure Patients

PONE-D-25-12369R2

Dear Dr. Mendonça dos Santos,

We’re pleased to inform you that your manuscript has been judged scientifically suitable for publication and will be formally accepted for publication once it meets all outstanding technical requirements.

Kind regards,

Chiara Lazzeri

Academic Editor

PLOS One

Additional Editor Comments (optional):

Reviewers' comments:

Reviewer's Responses to Questions

**Comments to the Author**

Reviewer #1: All comments have been addressed

2. Is the manuscript technically sound, and do the data support the conclusions?

Reviewer #1: Yes

3. Has the statistical analysis been performed appropriately and rigorously?

Reviewer #1: Yes

4. Have the authors made all data underlying the findings in their manuscript fully available?

Reviewer #1: Yes

5. Is the manuscript presented in an intelligible fashion and written in standard English?

Reviewer #1: Yes

Reviewer #1: all comments have been fullfilled and authors should be complimented for performing a such high quality revision

**Do you want your identity to be public for this peer review?** For information about this choice, including consent withdrawal, please see our Privacy Policy

Reviewer #1: **Yes:** Fabrizio D'Ascenzo

---

## [Editor Report · Acceptance letter]

PONE-D-25-12369R2

PLOS One

Dear Dr. Mendonça dos Santos,

I'm pleased to inform you that your manuscript has been deemed suitable for publication in PLOS One. Congratulations! Your manuscript is now being handed over to our production team.

Kind regards,

on behalf of

Dr. Chiara Lazzeri

Academic Editor

PLOS One